# Tracking the extensive three-dimensional motion of single ions by an engineered point-spread function

Yong-zhuang Zhou [1,5], Man-chao Zhang[1,2,3,5], Wen-bo Su [1,3], Chun-wang Wu [1,3], Yi Xie[1,3], Ting Chen[1,3], Wei Wu [1,3,4], Ping-xing Chen [1,3,4] ✉ & Jie Zhang [1,3] ✉

Three-dimensional (3D) imaging of individual atoms is a critical tool for discovering new physical phenomena and developing new technologies in microscopic systems. However, the current single-atom-resolved 3D imaging methods are limited to static circumstances or a shallow detection range. Here, we demonstrate a generic dynamic 3D imaging method to track the extensive motion of single ions by exploiting the engineered point-spread function (PSF). We show that the image of a single ion can be engineered into a helical PSF, thus enabling single-snapshot acquisition of the position information of the ion in the trap. A preliminary application of this technique is demonstrated by recording the 3D motion trajectory of a single trapped ion and reconstructing the 3D dynamical configuration transition between the zig and zag structures of a 5-ion crystal. This work opens the path for studies on single-atom-resolved dynamics in both trapped-ion and neutral-atom systems.

High-precision single-atom-resolved imaging is an essential technique for extracting the details of atomic systems. Currently, two-dimensional imaging of single atoms is well established in many applications, such as the determination of the position of a single ion[1], the resolving of a single neutral atom in a two-dimensional optical lattice[2–4], the imaging of trapped ions with high resolution[5–7] and the entanglement of neutral atoms via local spin exchange[8]. A few 3D imaging techniques exist that can determine the 3D positions of single atoms[9–15] and trapped ions[16,17]. However, these 3D imaging methods are limited to observing single atoms in a static state or in a shallow depth range. For the study of the dynamic properties of single atoms, precise 3D motion imaging of single atoms is needed; some examples include observing the spatial transportation and quantum tunneling of ion qubits between different ion chips[18], investigating the complex structural phase transition of trapped ion crystals[19,20], detecting the dynamics of single elementary or hybrid atomic systems[21–25], and monitoring defects and the construction process of 3D atom arrays[26].

However, until this study, 3D dynamic tracking for single atoms has remained elusive.

Due to the limited achievable detection range and inaccurate determination of the position along the direction of the optical axis of the objective lens, it remains challenging to extend conventional 3D imaging methods to detect the extensive dynamic motion of single atoms. In the direction of the optical axis, the atom's position is commonly obtained by measuring the size of the out-of-focus Airy disk[16,27,28], which leads to three primary drawbacks. First, the images obtained with positive defocus and negative defocus appear similar; hence, the sign of defocus can not be determined. Second, the image near the focal plane contains minimal information regarding the position of the atoms along the optical axis, which limits the measurement precision. Third, a large measurement uncertainty emerges as the amount of defocus increases; this causes a serious problem for the determination of 3D position at a large defocus. The location of single atoms along the optical axis can also be precisely determined by

[1]Institute for Quantum Science and Technology, College of Science, National University of Defense Technology, Changsha 410073, China. [2]Northwest Institute of Nuclear Technology, Xi'an 710024, China. [3]Hunan Key Laboratory of Mechanism and Technology of Quantum Information, Changsha 410073, China. [4]Hefei National Laboratory, Hefei 230088, China. [5]These authors contributed equally: Yong-zhuang Zhou, Man-chao Zhang. ✉e-mail: pxchen@nudt.edu.cn; zj1589233@126.com

image-stacking techniques[11,26,29,30], but numerous scan steps along the optical axis are required to cover the entire detection range. For our aim of fast imaging the 3D motion of single atoms with a large detection range, conventional 3D imaging technology for single atoms can not provide sufficient support.

In this study, we propose and demonstrate the extensive 3D motion tracking of single ions by exploiting the engineered point-spread function (PSF), which has been shown to be an effective method for detecting the 3D position of biological molecules at the tens of nanometers scale[31,32]. In our demonstration, a phase mask is employed in a 4f relay imaging system, which leads to a helical PSF in the imaging process. Then, the ion's position along the optical axis of the objective is encoded in the orbital-momentum-induced rotation of the two closely spaced PSF lobes. By combining the signal-to-noise ratio of images, we show that 3D motion with a range of up to 20 μm can be tracked with a theoretical position sensitivity better than 20 nm, which can be further improved by using objectives with larger numerical apertures (NA). As a preliminary demonstration, this method is applied for the detection of the 3D forced vibration of a single trapped ion. As a result, a 3D trajectory in the Lissajous pattern is observed. To show the application of this technique, we also record the dynamic transition between the zig and zag spatial configurations of a 5-ion crystal. Furthermore, it is expected that other engineered PSFs can likely be utilized to address various specific needs of different motion detection applications in atomic systems.

## Results

### Imaging configurations

The imaging is based on collecting fluorescence photons emitted by $^{40}Ca^+$ ions trapped in a blade trap[33]. In addition to the conventional $xyz$ coordinate system by which the secular motion of ions is derived, we introduce a convenient $hvz$ coordinate system according to the directions of the trap axis ($z$) and optical axis ($v$) of the imaging system (Fig. 1a). The energy level diagram and transitions of $^{40}Ca^+$ are shown in Fig. 1b; here, two different resonant lasers are primarily used. One is the repumping laser at 866 nm resonant with the $3^2D_{3/2} \leftrightarrow 4^2P_{1/2}$ transition, and this laser is used to repump the population out of $3^2D_{3/2}$. The other is a cooling and detection laser at 397 nm, and this laser can optically drive the $4^2S_{1/2} \leftrightarrow 4^2P_{1/2}$ transition. Since the lifetime of $4^2P_{1/2}$ is only approximately 7.1 ns[34], the ion can continually emit fluorescence photons under the detection laser, and a photon count rate of 114 k/s can usually be reached in our system.

The 3D information of emitters can be encoded in the variation of their fluorescence images using engineered PSFs in the form of rotation[35–37], translation[38,39] or more complicated pattern changes[40]; among these, the helical PSF yields two compact bright lobes with a relatively large depth range, making it the most widely used technique in biomedical super-resolution imaging. In our study, we employ a 4f relay system to access the Fourier plane of the microscope, where a phase mask is placed to generate the helical PSF with the aim of encoding the axial position of the ion via two-lobe rotation. The resulting PSF can be expressed by the following:

$$I_{PSF}(z, h, v) \propto \left| \mathcal{FT}\left\{ A(\rho, \phi) e^{2\pi i (\psi(\rho, \phi) + D(\rho, v))} \right\} \right|^2, \quad (1)$$

where $(\rho, \phi)$ are the normalized pupil coordinates and $A(\rho, \phi)$ is the pupil amplitude. $D(\rho, v)$ is the defocus phase that occurs when the emitter is displaced by $v$ from the focal plane, which can be described by the Hanser model[41] or the Gibson-Lanni model[42] depending on whether the objective is located inside or outside of the vacuum

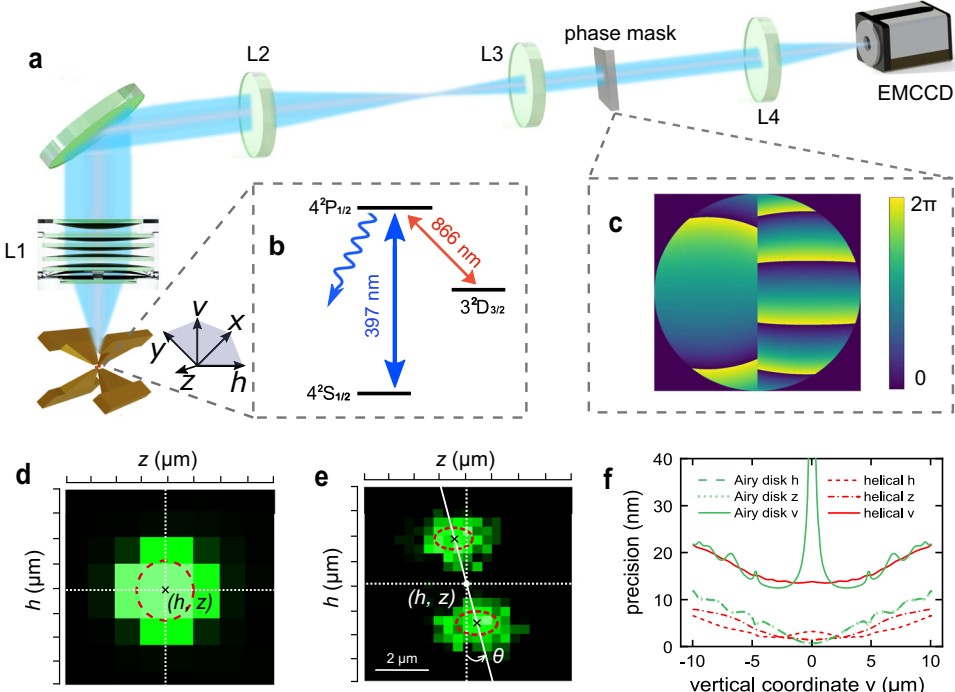

**Fig. 1 | Experimental scheme and setup. a** Imaging system design to capture the ion's position. The fluorescence emission of the ion trapped in a blade trap is collected by the objective lens L1 with a nominal NA of 0.4, and an intermediate image is created by the tube lens L2. This intermediate image is relayed to an electron-multiplying charge-coupled device (EMCCD) by a 4f imaging system to result in an effective magnification of 35.7. The phase mask is located at the confocal point of lenses L3 and L4. **b** Imaging-related energy levels and transitions of $^{40}Ca^+$. **c** Phase profile for PSF modulation. **d** Image of a single ion without the helical PSF modulation. We are able to determine the ion's centroid position in the $h$ and $z$ directions by fitting it to a 2D Gaussian function. False color represents the grayscale in the image. **e** Image of a single ion after helical PSF modulation. A 2D two-peak Gaussian fitting model can be employed to extract the centroid data $(h, z)$, while the displacement information along optical axis is mapped into the angle $θ$. **f** Theoretical comparison between the localization precision of the helical PSF (red curves) and defocused Airy disk (green curves) methods estimated using the Cramer-Rao bound metric under same imaging conditions (0.4 NA, 22k signal photons per ion and 10 background photons per pixel). The precision along $v, h, z$ axes are reflected by the solid, dash, and dot-dash curves respectively.

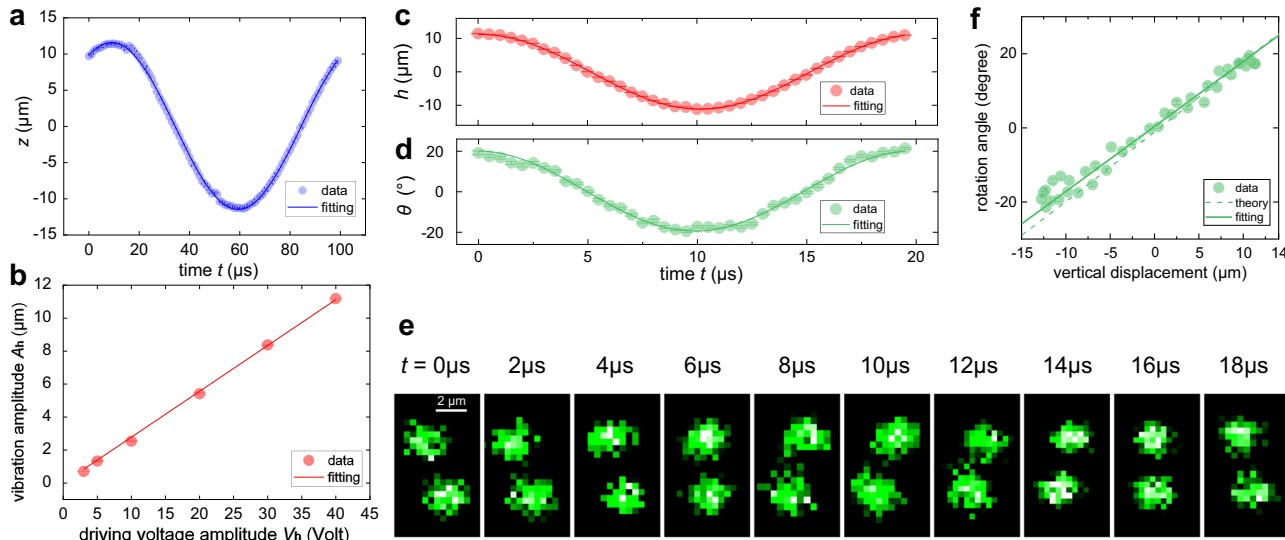

**Fig. 2 | Sinusoidal forced vibration and imaging system calibration. a** Sinusoidal forced vibration along $z$ direction when driven by a designed RF signal, with $\nu_z = (2\pi)128$ kHz, $\omega_z = (2\pi)10$ kHz, and $V_z = 1$ V. **b** The $h$ axis oscillation amplitude $A_h$ as a function of driving strength $V_h$. The frequency of secular motion and driving signal are $\nu_r = (2\pi)670$ kHz and $\omega_h = (2\pi)50$ kHz, respectively. The response coefficient is fitted as $\alpha_h = 2.06 \pm 0.03$ m$^{-1}$. **c** Detection of one-dimensional harmonic vibration along $h$ direction at different times. **d** The angle $\theta$ as a function of time $t$ when the ion is driven to oscillate along the $\upsilon$ axis. **e** Images of the moving ion along $\upsilon$ axis in ten time bins with duration $\Delta t = 0.5$ μs separated by 2 μs each. False color represents the gray scale in the image. The experimental conditions of **c**–**e** are kept the same, and related parameters are set as $\nu_r = (2\pi)670$ kHz, $V_h = V_\upsilon = 40$ V, $\omega_h = \omega_\upsilon = (2\pi)50$ kHz. Error bars are standard deviations from image fitting, scale bar corresponds to 2 μm in the object space. **f** Comparison of experimental data (same dataset as **c**, **d**), simulation, and fitting for the rotation angle as a function of vertical displacement $\upsilon$.

chamber. $\psi(\rho, \phi)$ is the phase retardation term introduced by the phase mask, which can be implemented by a refractive laser-written mask on a fused-silica substrate or using a liquid-crystal spatial light modulator.

The phase retardation pattern for a helical PSF can be generated using two methods: the Gauss-Laguerre-mode-based method[37,43] or the Fresnel-zone-based method[44]. We employ the latter and it is defined on circular zones in the Fourier plane as follows:

$$\psi_0(\rho, \phi) = (2l - 1)\phi, \ \left(\tfrac{l-1}{L}\right)^\eta < \rho < \left(\tfrac{l}{L}\right)^\eta, \ l = 1, \cdots, L, \quad (2)$$

where $L$ is the number of angular Fresnel zones with $l$ referring to the $l$-th zone. $\eta$ is a parameter that determines the peak confinement and shape invariance of the helical PSF during rotation[44]. Figure 1c shows an example phase pattern used in our experiment; such patterns originate from $\psi_0$ (e.g., $L = 4$, $\eta = 0.9$ for a high peak confinement and fast rotation rate) and are optimized using an adapted version of the Gerchberg-Saxton algorithm, where the light field in the image space is sampled along the optical axis to boost the two lobes at different rotation angles and to reduce the number of diffraction edges[37,44] (see Supplementary Note 4 for more details of the phase mask generation and implementation). When such a phase mask is implemented, a typical Airy disk image (Fig. 1d) of the single ion is decomposed into two separated lobes, and each corresponds to half of the phase modulation pattern in Fig. 1c. This phase modulation introduces orbital angular momentum to the light field, leading to light field rotation with propagation: When the emitter is displaced in the $\upsilon$ direction from the focal plane, the additional phase term (i.e., $D(\rho, \upsilon)$ in Eq. (1)) results in a continuous rotation of the two bright lobes in the PSF around their mutual center (Fig. 1e). By fitting the raw images to a 2-dimensional (2D) two-peak Gaussian function, the $h$ and $z$ coordinates of the ion can be derived by averaging the centroid data of two lobes, and the $\upsilon$ coordinate can be deduced from the rotation angle of the PSF (see Methods).

The Cramer-Rao lower bound (CRLB) metric is used to estimate the theoretical localization precision assuming a shot-noise limited model[45]. The CRLB, being the inverse of the Fisher information, represents the best/smallest possible variance of the parameters that can be reached when estimating the position of an ion, regardless of the estimator[46]. Thus, the square root of the CRLB indicates the theoretically best localization precision. For an average of 22k collected signal photons per ion and 10 background photons per pixel (corresponding to that achievable in experiment), our PSF design can possibly yield a 3D localization precision better than 20 nm over a depth range of 20 μm (Fig. 1f). Compared to the commonly used method of defocused Airy disk where various spot sizes are employed to deduce the depth, the helical PSF demonstrates the following advantages: (1) it exhibits much better axial sensitivity (i.e., along $\upsilon$) near the focus; (2) there is no ambiguity in determining the sign of defocus; this provides more than double the detection range and an-order-of-magnitude better range-to-precision ratio (defined as the detected range divided by the average precision over the range in the vertical direction). (3) The theoretical lateral precision (i.e., in $z$ or $h$) is also better at large defocus. The amount of Fisher information in the PSF image, regarding the ion's axial coordinate $\upsilon$, is positively correlated with its variation rate in this direction, i.e., $\frac{\partial I_{PSF}}{\partial \upsilon}$[45]. The Airy disk yields no intensity variations along the optical axis direction near the focal plane, leading to its poor localization precision in $\upsilon$. In addition, engineered PSFs enable a definable depth range by altering the phase parameters[44] or even the phase mask types[47]; thus, it is suitable for various application scenarios. Comparison of different phase parameters and imaging conditions can be found in Supplementary Note 5.

## Forced vibrations of a single ion

To track the large motion range of a single ion, we drive the ion with extra RF electrical signals to excite forced vibrations[48,49] since the amplitude of motion after Doppler cooling is less than 200 nm. Near the geometrical center of the trap, two pairs of additional electrodes

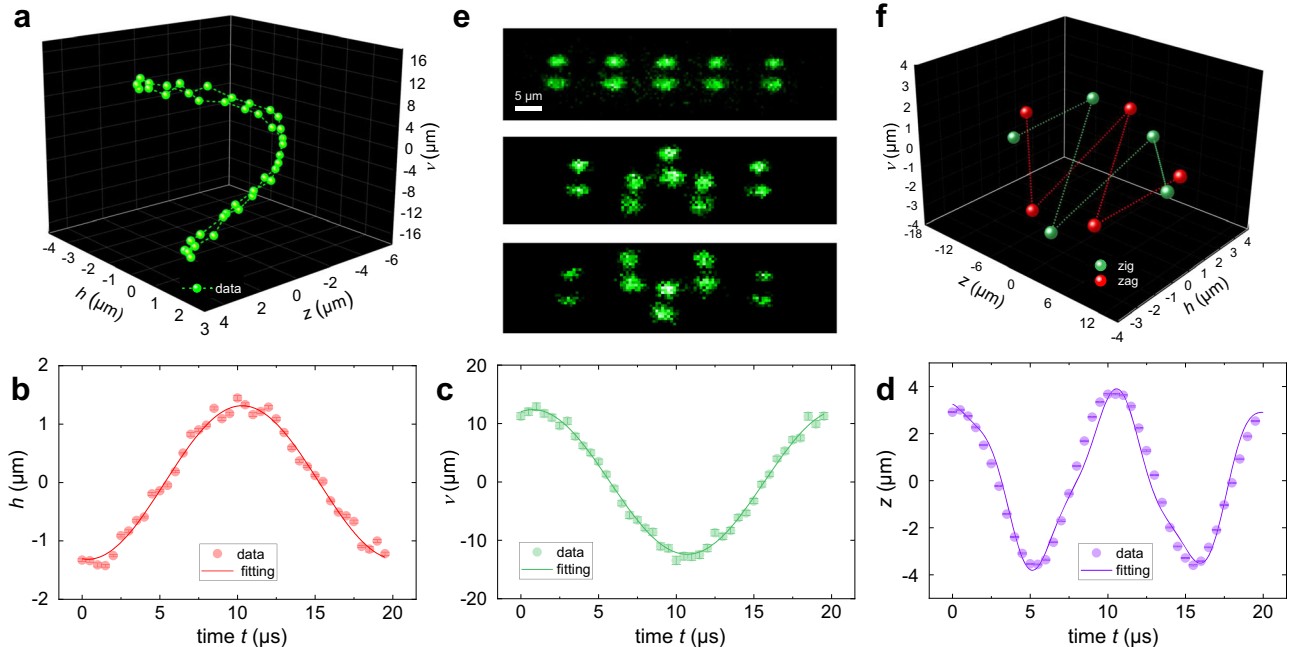

**Fig. 3 | Observation of ion's 3D motion trajectory and structural phase transitions in a 5-ion crystal. a** Reconstructed 3D motion trajectory. Its projected trajectories on $h, v, z$ axes with fitting curves based on theoretical model Eq. (3) and (4) are shown in **b**–**d**, respectively. Experimental parameters are $\nu_z = (2\pi)236$ kHz, $\omega_z = (2\pi)100$ kHz, $V_z = 1$ V, $\nu_r = (2\pi)670$ kHz, $\omega_h = \omega_v = (2\pi)50$ kHz, $V_h = 5$ V and $V_v = 40$ V. **e** The raw photography of linearly configured 5-ion crystal and zig, zag configurations. False color represents the gray scale in the image. **f** Reconstructed 3D configurations of zig (red dots) and zag (green dots) modes.

are located 3.85 mm away from the ion to compensate for stray electrostatic fields. Therefore, we can apply driving voltages in the form of $F_i = V_i \cos(\omega_i t), i = h, v, z$ on these electrodes as well as one of the tips to deliberately impose external driving forces on the ion along the $h, v$, and $z$ directions. During the driving process, the cooling lasers are turned off to avoid motion dissipation and unwanted photon collection. In the off-resonant condition, i.e., when the driving frequency is smaller than the secular oscillation frequency $\nu$, an extra ion-shift RF signal at ultralow frequency is also applied just before the driving signal to suppress the high-frequency modulation from secular motion (see Methods). Define $\nu_r = \nu_h = \nu_v$ if $\omega_i \ll \nu_i$, then the forced vibration of the trapped single ion can be written as follows:

$$i(t) = \frac{e\alpha_i V_i}{m\nu_i^2}\cos(\omega_i t), i = h, v, z, \tag{3}$$

where $\alpha_i$ is a constant representing the response coefficient of an ion to $F_i$. $e$ and $m$ denote the charge and mass of a $^{40}Ca^+$ ion, respectively. Eq. (3) indicates that when $F_i$ is implemented, the ion experiences a steady-state oscillation, and its motion amplitude is proportional to the strength $V_i$ of the driving signal. To characterize the quality of the designed sinusoidal forced vibration, we investigate the axial ($z$) oscillation (Fig. 2a) by scanning the delay time $t$ before the detection process in each oscillation period while fixing the detection pulse duration to $\Delta t = 1$ µs; this duration is much shorter than the secular motion period of $2\pi/\nu_z = 7.81$ µs (see Methods). By fitting each image to obtain the 3D coordinates at the different times $t$, the ion trajectory is reconstructed. The data points reveal negligible modulation from $\nu_z$ and can be precisely fitted by a cosine function at frequency $\omega_z$. The phase shift in Fig. 2a mainly results from the electronics used in the experimental setup including capacitors and inductors. In addition, the ion motion amplitudes $A_h$ under different driving voltages $V_h$ are also investigated, as shown in Fig. 2b. The result shows a linear dependence of $A_h$ on $V_h$, as expected in Eq. (3).

To attain 3D tracking of ion motion, precise calibration of the imaging system along the $v$ direction is crucial. For this purpose, we measure the one-dimensional harmonic vibration of a trapped ion along $h$ and $v$, as shown in Fig. 2c–e. The secular motional modes in two radial directions are almost degenerate in our trap (less than 0.5% difference in trap frequency); if the same driving voltages are applied, then the forced vibrations in the two scenarios should exhibit the same amplitude and period. Because the $h$ motion is directly mapped to the lateral displacement of the image, we can use the $h$ motion as a reference to calibrate the $v$ motion (defocus) of the ion. By analyzing the displacement along the $h$ axis with a cosine function, we obtain a motion amplitude of $h_a = 11.17 \pm 0.05$ µm under a driving signal with $V_h = 40$ V and $\omega_h = (2\pi)50$ kHz. Similar data processing is also implemented for the angle $\theta(t)$, and the amplitude is determined to be $(19.76 \pm 0.43)°$. Due to the installation error of the phase mask, an offset angle $(0.39 \pm 0.55)°$ is also obtained in this fitting. Then, the fitting constants $\beta, c$ that reveal the relationship between the rotation angle $\theta$ and the vertical displacement $v$ can be found in the linear model $v(\theta) = \beta\theta + c$, and the parameters are calibrated as $\beta = \frac{180}{\pi} \times (0.57 \pm 0.02)$µm/rad and $c = -0.22 \pm 0.32$ µm in Fig. 2f. The theoretical simulation result is also shown in Fig. 2f, which is consistent with the calibration data. Note that a slight difference exists because many factors could compromise the nominal imaging performance. For example, the trap blades, the vacuum view window, or the phase mask can introduce pupil cropping and wavefront aberrations. Considering the weak fluorescence and the inaccessibility of the vacuum chamber, other types of aberrations may be introduced due to the difficulty in aligning the optics. For this reason, pre-calibration is vital for such systems to account for existing systematic errors.

Figure 3a–d show the reconstructed 3D trajectory of a single ion's motion (see Supplementary Movie 1 for a dynamic illustration). We force the ion to undergo harmonic vibration in all three directions by simultaneously applying three RF driving signals. The ion trajectory forms an '8'-shaped Lissajous pattern in 3D space as expected (Fig. 3a), and its projections on the $h, v$, and $z$ axes are shown in Fig. 3b–d,

respectively. Due to $\omega_{h,v} \ll \nu_r$, the oscillations along the $h$ and $v$ axes can both be modeled by a simple cosine function with different amplitudes and an additional delayed phase. The motion along the $z$ axis, however, is more complicated. Owing to $\omega_z = (2\pi)100$ kHz and $\nu_z = (2\pi)236$ kHz, application of the $\omega_z \ll \nu_z$ condition to Eq. (3) is no longer valid. Therefore, a more generalized equation is required to describe the motion and it can be written as follows:

$$z(t) = \frac{e\alpha_z V_z}{m} \frac{\cos(\omega_z t) - \left(\frac{\omega_z}{\nu_z}\right)^2 \cos(\nu_z t)}{\nu_z^2 - \omega_z^2}. \tag{4}$$

This equation is used in the trajectory analysis along the $z$ axis, as shown in Fig. 3d, by introducing an extra delay phase (see Supplementary Note 1 for details). Instead of $\omega_z$, Eq. (4) indicates a complicated vibration with a longer motion period, which introduces errors into our scheme of signal acquisition (see Supplementary Note 2). As a result, the deviation between the experimental data and fitting curve in Fig. 3d is slightly larger than that in Fig. 3b, c.

### Structural phase transition in a Coulomb crystal

The zig and zag structures of the trapped ions are typical and experimentally available configurations, which are ideal candidates for investigating fundamental physics including non-equilibrium dynamics[50], and for demonstrating the application of this 3D motion imaging technique. Here, the 'zigzag' transition in an ion string of five $^{40}Ca^+$ ions is observed in our experiment by recording the dynamic evolution of the ion position. After loading five linearly configured ions (see the first row in Fig. 3e), by ramping the axial trap frequency $\nu_z$, the ions are initially squeezed into the zig or zag state (the second and third row), and the ions are theoretically clustered in the $xoz$ or $yoz$ plane due to the slight difference between the frequencies of two orthogonal radial modes[51]. Then, keeping the trap frequencies unchanged, we observe a transition between the two configurations under the influence of thermal fluctuations (see Methods). Figure 3e shows two raw images of the ions at two separate times, by which the 3D configurations of the zig and zag states are reconstructed, as shown in Fig. 3f. One can observe that the ions lie in an inclined 2D plane, and the spatial configuration transitions between 'W'-shaped and 'M'-shaped over time as expected (see Supplementary Movie 2 for a dynamic illustration of the structural phase transitions). Since the ion coordinates can be deduced from a single snapshot at one moment, our proposed approach is capable of monitoring the 3D position dynamics during the phase transition at a speed as fast as the frame rate of the camera.

## Discussion

In summary, we demonstrated a general dynamic motion imaging method at the atomic scale and achieved 3D motion tracking of a single trapped ion with an oscillation amplitude >10 μm, which has never been observed previously in cold ion or atom systems. By encoding the $v$-displacement in the angular orientation of two bright lobes, the location of ions with a precision better than 20 nm within the 20 μm defocus range can theoretically be achieved. Here, we stress that the imaging method demonstrated is universal for both trapped ions and neutral atoms, and the implementation of the detection scheme requires only a 4$f$ relay system and an appropriate phase mask. It is expected that future improvements in the imaging configurations (e.g., using 4$\pi$ microscope[52]), the NA (e.g., with custom-designed high NA objective[3]), the phase mask design (e.g., with the information-rich tetrapod PSF[53]), and the stability of the imaging system can lead to a theoretical spatial resolution below 10 nm, enabling the tracking of the flexible motion types (see Supplementary Note 5 for precision variations with respect to NA). Notably, real-time tracking of particles can be attained if a faster and more sensitive camera is used; this would

enable applications in the study of nonperiodic motion tracking. A vacuum-compatible piezo stage can be implemented within the vacuum chamber for direct calibration by capturing the image stacks, and the phase retrieval algorithms can then be employed to calculate and compensate for any existing aberrations. With the proof-of-principle experiments reported here, our approach opens a path to the study of 3D micromotion compensation[54] and spatial ion transportation in trapped ion systems[18]. In particular, this method will also enable direct and in-depth investigation of the complex structural phase transitions[19,20], the sensing of weak physical quantities with cold atoms[16,55], the observation of multibody dynamics in systems composed of single or multiple atomic species[21-25], and the detection of the quantum tunneling on ion chips and in 3D optical lattices[56].

## Methods

### Setup of forced vibrations

To create a large 3D motion range for the ion in the trap, three driving AC signals are applied to the compensation and endcap electrodes by exploiting custom-made combiners; these combiners are composed of two low-pass filters, one RC filter and one LC filter, to combine the DC voltage and driving signals with the least mutual interference (see Supplementary Note 1 for details). A cosine-type forced vibration is then formed by selecting a driving frequency less than half of the secular frequency and an ion-shift signal with an adiabatic ramping time of 500 μs; these settings are used to suppress the modulation induced by the secular motion and shift the ion to its maximum displacement adiabatically (see Supplementary Note 1 for details).

### Fluorescence collection for a single ion

The trajectory of the ion is monitored by collecting the fluorescence using an EMCCD (Andor Ultra 888). In principle, the movement trajectory recovered would be more precise with more sampling points, however, more sampling points indicates lower tracked motion frequencies. In order to balance this conflict, more than 10 sampling points are required to trace the motion in our experiment. However, the exposure time of the EMCCD is limited to 10 μs, and it is impossible to detect the motion with oscillation frequencies >10 kHz. To overcome this dilemma, we exploit the periodic motion of the ion by constantly exposing the EMCCD, while the 397 nm detection laser is applied periodically with a fixed time delay and a short duration $\Delta t = 1$ μs). By taking advantage of this fluorescence collection method, the time resolution problem of the EMCCD can be avoided and the detected motion frequency can be extended up to 100 kHz. Based on the above settings, we can collect an average of 22,000 photons to identify the ion position with 3000 iterations. Finally the position for each time delay can be obtained by accumulating all the photons collected in each iteration and the full motion can be traced by varying the delay time for the whole period (see Supplementary Note 2 for details). Note that on top of the light intensity variation rate, the standard deviation estimating the emitter's lateral coordinates scales with the inverse square root of the number of detected photons, assuming background-free imaging conditions[45]; thus, a $\sqrt{2}$ times better localization precision can be expected with doubled photon accumulation time.

### Transition of zig and zag configuration

The linear 5-ion string is stored in a trap with radial and axial trap frequencies of $(2\pi)973$ kHz and $(2\pi)300$ kHz, respectively. By ramping up the axial trap frequency, the linear ion string undergoes a structural phase transition and transforms into a zig or zig configuration[50]. Due to the slight difference (approximately 0.5%) between the two radial modes, the zigzag configuration is limited to a 2D plane. Since the ions are only cooled by Doppler cooling, the thermal fluctuations induce a dynamic structural transition between the zig and the zag states.

 

## Data analysis

The acquired images are first smoothed using a median filter, then a two-peak two-dimensional Gaussian model with 12 parameters are used to fit the PSF images, with the parameters being the amplitude $(A_1, A_2)$, center $(C_1(z_1, h_1), C_2(z_2, h_2))$, width $(\sigma_{1a}, \sigma_{1b}, \sigma_{2a}, \sigma_{2b})$ and orientation $(\gamma_1, \gamma_2)$ for both the two peaks; thus, the $z$ and $h$ coordinates can be deduced as $(z_1 + z_2)/2$ and $(h_1 + h_2)/2$. If we define the angle $\theta = \arccos(\overrightarrow{C_1 C_2} \cdot (0,1) / \parallel \overrightarrow{C_1 C_2} \parallel)$ between negative $h$ axis and the line connecting the fitted centers of two lobes, then a displacement parallel to $v$ axis can be revealed by a change in $\theta$ : $v(t) = \beta\theta(t) + c$, with $\beta$ and $c$ being constants within a specified defocus range.

## Data availability

The data that supports the findings are presented in the article and Supplementary Information. Source data are provided with this paper.

## Code availability

The code used for data analysis and simulation is available from the corresponding author upon request.

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

## Acknowledgements

We thank Professor Paul C. Haljan for the helpful discussion. This work is supported by the National Natural Science Foundation of China under grant no. 62105368 (Y.Z.), no. 12004430 (J.Z.), no. 12074433 (P.C.), no. 12174447 (W.W.), no. 12204543 (T.C.), no. 12174448 (C.W.), no. U2241288 (M.Z.) and the Innovation Program for Quantum Science and Technology under Grant no. 2021ZD0301605 (P.C.). J.Z. and Y.Z. acknowledge the support from the Science and Technology Innovation Program of Hunan Province under Grant No. 2022RC1194 and 2023RC3010. J.Z. acknowledges the support from the Natural Science Foundation of Hunan province under Grant No. 2023JJ10052.

## Author contributions

Y.Z. and J.Z. proposed the idea. J.Z., Y.Z., M.Z., and W.S. contributed to the theoretical analysis, design and setting up of the experiments. M.Z., Y.Z., J.Z., and W.S. performed the atom experiment and analyzed the data. W.W. and P.C. supervised the project. M.Z., Y.Z., and J.Z. wrote the original paper with input from C.W., Y.X., T.C., W.W., and P.C. All authors contributed to the revision of the manuscript.

## Competing interests

The authors declare no competing interests.
