## [Peer Review File · Nature Communications]

Tracking the extensive three-dimensional motion of single ions by an engineered point-spread functionREVIEWER COMMENTS

Reviewer #1 (Remarks to the Author):

Review of "Tracking the extensive three-dimensional motion of single atoms by engineered point-spread function"

Key Results

This paper describes a method for performing dynamic 3D imaging of an ion using a point-spread function that strikes out a double-helix in 3D space (as a function of defocus of the imaging system). This engineered PSF is inspired by similar work in the imaging of biological molecules. This is used to map out the motion of trapped ions, both a single ion and a crystal of multiple ions. The experimental system is presented along with experimental data showing that this motion can be accurately mapped with good time resolution. I particularly found the imaging method to be one of the more interesting parts of the paper and feel that it perhaps deserves promotion to the main text.

Validity and significance

The experimental data presented in this paper are nice, and the imaging method is, in my opinion, quite clever. I have no doubt that the data is valid and the conclusions are reasonable. The method presented here is of interest to the field and will likely catch on in similar experiments in atoms and ions. This is, however, a paper that is heavy on methodology and light on actual physics, with a mention of a physically-relevant phase transition between a zig and zag motion (which I admittedly know very little about, as this is not my field) seeming almost as a footnote.

Issues

This paper, however, has a number of issues that need to be clarified and/or rectified before I recommend publication. I outline these below:

--The word "atom" and "ion" are conflated in the paper in places. Don't use the word "atom" to refer to an ion.

--The paper is missing a number of relevant citations:

---When discussing methods for determining the 3D positions of neutral atoms, the absence of <https://journals.aps.org/prapplied/abstract/10.1103/PhysRevApplied.15.064047> is notable, as this method is particularly clever (and I suspect it's also applicable to dynamic systems). Likewise I think <https://opg.optica.org/ao/fulltext.cfm?uri=ao-56-31-8738&id=376141> should be noted; again, this method is likely usable with dynamic systems. It's likely there are more such methods out there; the authors should do a more thorough literature review.

---The image stacking method citations misses <https://journals.aps.org/prapplied/abstract/10.1103/PhysRevA.102.053311>

---The mention of an iterative Fourier transform method is missing a citation. How is this different from the common Gerchberg-Saxton algorithm?

---A reference to a paper that describes the blade trap (and the experimental apparatus as a whole, or a similar one) in greater detail would be very useful for the reader who is not an expert in ion trapping and manipulation.

--I find that details of the engineered PSF are lacking (and I am also not sure I would call it "elaborate"—it's actually pretty straightforward once you understand the underlying optics). There is

no background that gives the reader necessary details in terms of which specific phase mask was used to engineer the PSF. This should be illuminated, especially in regards to Figure 1(f). You present results for two different PSFs without telling us what they are and how good they should be. The analytics for this have been presented in a number of papers, e.g., <https://journals.aps.org/pre/abstract/10.1103/PhysRevE.54.R50>.

--On a related note: can't you get knowledge of your vertical position from knowledge of your PSF analytically? That is, you should be able to model analytically, especially for your NA, how your rotation angle should vary with v . How does this analytical model agree with your data? This should be done and presented alongside the data.

--Furthermore, how do you know you're saturating the Cramer-Rao bound? This metric seems, as a lower bound, is likely making your precision out to be better than it actually is, unless you can show that you're indeed hitting that bound. How are you determining the data shown in Figure 1(f)?

--What do the lines in Fig. 2 (and Fig. 3) mean—are they fits? How do they compare to what you expect analytically (i.e. ,how do you know that aberrations aren't causing you to misinterpret your data)? What do the error bars mean? The timestamps above Fig. 2(e) are sloppy and should be modified.

--What is an "extra delay phase"? Why is this important?

--You mention being able to image an ion's position in 3D in one snapshot, then you say that you can image phase transitions at a speed as fast as the camera's frame rate. Don't you need multiple images to determine phase transitions? This is confusing as written. It also seems somewhat arbitrary with regards to what the authors call the zig and zag modes. Is this common parlance in the field? I'm not sure here. If so, a citation to similar work that discusses this would be prudent.

--The authors should comment on how tightly packed the imaged atoms/ions can be before the two helicoidal points of adjacent emitters overlap and can thus cause confusion.

--I would like to see more details on how the phase mask is produced and implemented.

--Finally, while this paper presents some interesting work, but it is generally poorly written. This causes it to be confusing in parts. Extensive editing work is required before it will be acceptable for publication in a Nature journal, I believe.

Reviewer #2 (Remarks to the Author):

The authors of the manuscript "Tracking the extensive three-dimensional motion of single atoms by engineered point spread function" devise and demonstrate a novel imaging method to determine the three-dimensional position of trapped ions (or other point-like scatterers) under a microscope. Drawing on methods developed in other contexts, they implement a phase mask in the imaging path that transforms a Gaussian point spread function into a two-lobed structure, whose rotation encodes the axial position of the scatterer. Using this technique, they demonstrate the three-dimensional tracking of a single ion undergoing periodic driven motion and the zig/zag orientation of a five-ion crystal.

The imaging method demonstrated here is not completely novel conceptually, but to my knowledge, this is the first implementation in the context of quantum sciences and imaging of trapped atoms. The applicability of the scheme in most typical ion trapping scenarios is not quite clear, in particular, because the photon number requirements are not discussed very clearly (see below).

Nevertheless, the use of the helical PSF in this context is novel and may lead to interesting applications in ion or neutral atom trapping. The manuscript is clear, cites prior work appropriately, and described the data in sufficient detail.

I recommend publication of the manuscript provided that the authors can address the questions below.

There are two main questions that I feel are not discussed extensively enough in the main text.

1. Photon budget. As mentioned briefly in the methods, the resolution of any imaging method is ultimately limited by the number of detected photons. This fundamental limitation is not discussed at all in the main text and in my opinion, deserves a better exposition. In particular:

a. Looking at Figure 1F, the fundamental difference between naïve Gaussian and the helical PSF methods is not very large, especially in the transverse directions. Does the helical PSF actually outperform traditional methods once the finite efficiencies of the phase mask are taken into account?

b. The data shown here is for very large photon numbers of 22k photons per atom. This number is quite atypical for "normal" ion trap experiments, where high-fidelity imaging can be performed with significantly fewer photons. Does the helical PSF advantage translate to lower photon number of several hundred in more typical experiments?

2. Design of the helical PSF. It would be nice if the reader could build some intuition for the design of the helical point spread function and follow the considerations that went into the choice of this particular phase mask.

a. What is this phase mask, e.g. which Zernike polynomials contribute to the image shown in Figure 2c?

b. How do we understand the difference in transverse performance between the Gaussian and helical PSF displayed in Figure 1F? I assume the comparison is carried out at a fixed number of detected photons, but please clarify.

One could determine the position of an atom simply from the center of mass of all detected photons. In this case, I would imagine that at fixed photon number and resolution, a Gaussian PSF is the best choice because of its minimal spatial size.

If, on the other hand, the spatial position is determined from performing a more elaborate fit on the PSF, having sharper features (or more lobes) in the PSF might be preferable. Would having more lobes in the PSF improve the transverse resolution even further?

Please comment on the strategy of choosing a particular phase mask.

c. Is there experimental evidence that the theoretical improvement in resolution reported in Figure 1F is actually observed in the experiment?

Minor points

A. How is the phase mask implemented physically? What efficiency is observed for the physical implementation? How does that feed into the question of the photon budget mentioned above?

B. Line 60. Please specify how short the lifetime is and at what rate photons can be collected.

C. Line 78. How do the authors arrive at the "order of magnitude better range-to-precision ratio"? The

doubling of the range at roughly similar precision (except near the focus) would suggest this improvement is about a factor of two. If the focus is included, where the imprecision of the Gaussian method diverges, the factor would be infinity. Why one order of magnitude?

D. Line 113. The coefficients beta and c are not defined anywhere in the main text.

Reviewer #3 (Remarks to the Author):

This paper demonstrates a new approach to performing 3D imaging of single ions using a phase-mask to transform the point spread function from an Airy function to a helical beam profile that rotates proportional to the change in depth of focus, overcoming a major limitation of previous approaches to 3D imaging using either adjustable focal length lenses or moving optics, or by inferring from blurring and defocus of the Airy spots. To demonstrate the effect the authors show real-time imaging of a 5 ion crystal transforming between two different zig-zag configurations, and demonstrate a theoretical resolution of 20nm in a 20 μ m depth of focus. These results provide an interesting approach to performing 3D imaging which is relevant to work on both trapped ion systems and emerging neutral atom tweezer arrays, making this an important and impactful paper.

However, in current form, there is not sufficient detail to allow another group to easily adopt and replicate this data which makes it hard to recommend for publication in Nature Comms in the current form despite the novelty of the results. For example, there are two pages of supplementary material explaining the ion trap setup and almost no detail at all about the design and optimisation or fabrication of the phase-mask that underpins the results which for most readers is the crucial part of the paper. Additionally, there is a lack of discussion around how this readout scheme can be used in the situation of high-fidelity readout and state detection which for many applications of 3D imaging is required.

I propose the authors resubmit a revised manuscript addressing the main criticisms below to allow consideration of whether this is sufficiently detailed for publication in Nature Comms.

(1) As mentioned above the key focus of this paper is on use of the phase-mask to transform the PSF, but there just is insufficient detail supplied. On page 2 it simply states the phase mask was "optimized using the iterative Fourier transform algorithm to reduce the number of diffraction edges." but this requires more detail so a reader could design a suitable phase mask from scratch e.g. what is the cost function used in the minimisation process? How was the initial phase-mask chosen to choose the mapping from focus to angle? There is also no detail about mask construction - how is the experimental phase mask fabricated, from what materials and with what process? What is the tolerance on the mask compared to the theoretical design?

(2) For performing high fidelity imaging, the aim is to obtain high collection efficiency onto a small group of pixels. For the helical PSF, how efficient is the transformation, and how would the fidelity of detection evolve comparing the Airy and helical PSF accounting for the change in number of pixels and larger contribution to background counts of spreading the signal over a larger number of pixels.

(3) The authors quote a resolution of 20 nm at 20 μ m depth of focus. How do these numbers scale with e.g. NA - if you increase the NA from 0.4 to 0.5 how does the depth of focus and resolution change- is it a simple relationship?

(4) For many of the example applications of imaging single atoms and ions another important factor is cross talk. How close can I resolve two objects at different distances? Is there a trade-off between the design of the helical and the resulting resolution that can be obtained? Ideally the figures should be clearly labelled with distance in μ m to aid in resolving the PSF size scale relative to atom spacing.

(5) In the abstract and manuscript the expression "a thin detection range" is used and this is not sufficiently clear - I believe what is meant is a shallow depth of focus but clarity is required in this phrasing.

(6) In the introduction where "3D methods for imaging" are discussed it would be helpful for the general reader to appropriately cite prior work performing 3D readout using movable lenses (e.g. <https://doi.org/10.1038/nphys645>), focus adjustable lenses (ref 22) and using an SLM to re-map the 3D space onto a 2D plane (<https://doi.org/10.1364/OE.415805>)

(7) Whilst the paper is generally well written, there remain many instances where articles are incorrectly used or the meaning is unclear e.g. in the abstract "acquisition of ion's 3D position" should be "the ion's position", and on line 33 "In direction, the position..." - what direction is being referred to? Please carefully review the grammar.

(8) An equation is given to describe motion (eq 2) with no citation - how is this derived?

I. RESPONSE LETTER TO REVIEWERS (NCOMMS-23-50566-T)

To all reviewers: We deeply appreciate the thorough review of our manuscript, which has led us to further improve our manuscript. After we carefully considered the three reports, we have made a major revision to the manuscript and addressed all reviewers' comments point-by-point with a list of revision (LOR) attached at the end of this letter. To help reviewers check the reply letter quickly, we make a brief summary of our replies for common concerns as follows.

1. The English of the whole manuscript has been revised carefully with the help of English editing service for better expression and the revised parts are marked in red.
2. More information regarding the generation of the helical PSF and its corresponding phase mask has been added to the main text.
3. Fig.2f is added to make a clear and direct comparison between the theoretical imaging model and the experimental data.
4. Detailed information of the imaging model, a flowchart for the phase mask optimization, a discussion on the phase mask fabrication and efficiency have been added as a new section IV in the supplement.
5. Experimental comparison between the conventional method of defocused Airy disk and the helical PSF have been added as a new section V of the supplement.
6. Theoretical comparison of the localization precision among scenarios of various numerical aperture and photon levels has also been added to section V of the supplement.

Reviewer #1 (Remarks to the Author):

Review of "Tracking the extensive three-dimensional motion of single atoms by engineered point-spread function"

Key Results

This paper describes a method for performing dynamic 3D imaging of an ion using a point-spread function that strikes out a double-helix in 3D space (as a function of defocus of the imaging system). This engineered PSF is inspired by similar work in the imaging of biological molecules. This is used to map out the motion of trapped ions, both a single ion and a crystal of multiple ions. The experimental system is presented along with experimental data showing that this motion can be accurately mapped with good time resolution. I particularly found the imaging method to be one of the more interesting parts of the paper and feel that it perhaps deserves promotion to the main text.

Response: Thanks for the valuable comment and positive support on this work. We agree with reviewer that the imaging method is of interesting for this paper and we feel sorry that we did not provided enough details in either main text or supplement. To make a more detailed and clear description of the imaging method, we have added the generation details of the helical PSF to the main text in line 65 to 78. We also note that the imaging method

with helical PSF is application specific, hence we provide more detailed information in the supplement, including the design flow, the imaging model, the implementation materials and a comparison of performance with various design parameters, hopefully we have enough information for this imaging method.

Validity and significance

The experimental data presented in this paper are nice, and the imaging method is, in my opinion, quite clever. I have no doubt that the data is valid and the conclusions are reasonable. The method presented here is of interest to the field and will likely catch on in similar experiments in atoms and ions. This is, however, a paper that is heavy on methodology and light on actual physics, with a mention of a physically-relevant phase transition between a zig and zag motion (which I admittedly know very little about, as this is not my field) seeming almost as a footnote.

Response: Thanks for the valuable comment on our manuscript, especially for the positive comment on our experimental data and conclusion. We agree with reviewer that this paper focused too much on the method part. After careful consideration of this opinion, we think the main purpose of this work is to introduce a new imaging method for detection of the dynamic motion of atoms so that this technique can stimulate some interesting fundamental physics such as the non-equilibrium physics. We notice that the zig and zag structures are common and easily available in trapped-ion system and the dynamics of these two structures is very suitable for verifying our 3D motion imaging technique, therefore we modified the sentences in lines from XX to XX to be “The zig and zag structures of trapped ion are typical and experimentally available configurations, which are ideal candidates for investigating the fundamental physics including non-equilibrium dynamics [?] and for demonstrating the application of this 3D motion imaging technique.” What’s more, the detection of the complex structural phase transitions (e.g. double helical and 3D configuration of Coulomb crystals with our method will be our next research topic.

Issues

This paper, however, has a number of issues that need to be clarified and/or rectified before I recommend publication. I outline these below:

–The word “atom” and “ion” are conflated in the paper in places. Don’t use the word “atom” to refer to an ion.

Response: Thanks for the comment and suggestion, we have revised the word “atom” to “ion” in appropriate places including the title, abstract and introduction, where we want to describe an ion instead of an atom (LOR1 and LOR2).

— The paper is missing a number of relevant citations:

—When discussing methods for determining the 3D positions of neutral atoms, the absence of <https://journals.aps.org/prapplied/abstract/10.1103/PhysRevApplied.15.064047> is notable, as this method is particularly clever (and I suspect it’s also applicable to dynamic systems). Likewise I think <https://opg.optica.org/ao/fulltext.cfm?uri=ao-56-31-8738&id=376141> should be noted; again, this method is likely usable with dynamic systems. It’s likely there are

more such methods out there; the authors should do a more thorough literature review.

Response: Thank you for reminding us the potential methods for dynamic imaging of cold atoms. The two papers [Phys. Rev. Applied, 15, 064047 (2021), Applied Optics, 56, 8738 (2017)] indeed have potential for dynamic imaging, we believe they will be used for the purpose in near future and we have added these references to line 24. After careful checking the three-dimensional imaging literature, we did find more related work on this topic [Optics Express, 3, 415805 (2021), Nature Mater 14, 1099–1103 (2015)], we have also cited these papers in our introduction part in line 24 of the main text (LOR4).

—The image stacking method citations misses <https://journals.aps.org/pr/abstract/10.1103/PhysRevA.102.053311>

Response: Thank you for the suggestions, we have added this paper to our reference in line 39 in the main text, the sentence is modified to be “...The location of single atoms along the optical axis can also be precisely determined by the image-stacking techniques [11, 26, 29, 30],...” and reference [30] is the paper Phys. Rev A, 102, 053311 (LOR4).

—The mention of an iterative Fourier transform method is missing a citation. How is this different from the common Gerchberg-Saxton algorithm ?

Response: Thanks for the comment. (i) We have added two citations [Opt. Express 16, 3484–3489 (2008), Opt. Express 26, 4873-4891 (2018)] for the iterative Fourier transform method in line 84.

(ii) We are very sorry that we did not state this clearly, the iterative Fourier transform method is “an adapted version of the Gerchberg-Saxton algorithm where the light field in the image space is sampled along the optical axis so as to boost the two lobes at different rotation angles and to reduce the number of diffraction edges.” We have now made this clearer in the main text in lines from 84 to 86 (LOR11), and have provided a flowchart (Fig.R1) for the phase optimization algorithm in the supplement (LOR22).

FIG. R1. Iterative algorithm to generate the helical phase mask.

—A reference to a paper that describes the blade trap (and the experimental apparatus as a whole, or a similar one) in greater detail would be very useful for the reader who is not an expert in ion trapping and manipulation.

Response: Many thanks for this valuable suggestion, we have added a PhD thesis as the reference to give details of the experimental apparatus in line 55 of main text (LOR5).

—I find that details of the engineered PSF are lacking (and I am also not sure I would call it “elaborate”—it’s actually pretty straightforward once you understand the underlying optics). There is no background that gives the reader necessary details in terms of which specific phase mask was used to engineer the PSF. This should be illuminated, especially in regards to Figure 1(f). You present results for two different PSFs without telling us what they are and how good they should be. The analytics for this have been presented in a number of papers, e.g., <https://journals.aps.org/pre/abstract/10.1103/PhysRevE.54.R50>.

Response: We thank valuable comment. We have provided the description of the engineered PSF for better context in lines from 65 to 68: “The 3D information of emitters can be coded in the variation of light field in the form of rotation [Phys. Rev. E 54, R50 (1996), Opt. Lett., 31, 181 (2006), Opt. Express 16, 3484 (2008)], translation [Nat. Photon. 8, 302 (2014), Phys. Rev. Lett., 124, 198104 (2020)] or more complicated pattern changes [Phys. Rev. Lett., 113, 133902 (2014)], among which the helical PSF yields compact bright lobes with a relatively large depth range, making it the most widely used technique in biomedical super-resolution imaging...” (LOR8). In addition, we provided details of the specific helical PSF and its corresponding phase mask design that we employed in this work in lines from 69 to 81 and in supplement Section IV. Now the description of the phase mask design in the main text is “...where a phase mask is placed to generate the helical PSF with the aim to code the ion’s axial position as two-lobe rotation. The resulting PSF can be expressed by...”, “...the phase pattern in our experiment, originating from ψ_0 ($L = 4, \eta = 0.9$ for a high peak confinement and fast rotation rate) and optimized using an adapted version of the Gerchberg-Saxton algorithm...” (LOR9, LOR10, LOR11)

We agree with reviewer and have deleted the inappropriate word “elaborate” in the main text.

In Fig.1f, we were comparing the conventional method of defocused Airy disk and the helical PSF of two different mask modulation strength, i.e. different L . We agree it was confusing with too many lines in one plot and not enough description in the text, therefore we have updated Fig.1f and revised corresponding description for this comparison in lines from 96 to 102. Besides, we have moved the comparison of various phase mask strength to the supplementary section IV with a more detailed description: “...A larger L leads to larger lobe spacing and slower angular rotation rate, and thus a larger operable depth range; theoretically, it yields a worse localization precision in the axial direction (Fig.S6)...”

—On a related note: can’t you get knowledge of your vertical position from knowledge of your PSF analytically? That is, you should be able to model analytically, especially for your NA, how your rotation angle should vary with

v. How does this analytical model agree with your data ? This should be done and presented alongside the data.

Response: We thank reviewer for this valuable comment. We have rigorously modeled our the imaging system and added a comparison of the data to theory as can be seen in Fig.R2. This is added as a new Fig.2f in the main text and we have added the following description in lines 141 to 145: “The theoretical simulation result is also shown in Fig.2f, which is consistent with the calibration data. Note that a slight difference exists, because many factors could compromise the nominal imaging performance. For example, the trap blades, the vacuum view window, or the mask can introduce pupil cropping and wavefront aberrations. Considering the weak fluorescence and the inaccessibility of the vacuum chamber, other types of aberrations may be introduced due to the difficulty in the alignment of optics. For this reason, pre-calibration is vital for such systems to account for existing systematic errors.” To completely describe the system, “A vacuum compatible piezo stage can be implemented within the vacuum chamber for direct calibration and acquisition of image stacks. Phase retrieval algorithms can then be employed to calculate and compensate any existing aberrations.” These added discussions can be found in lines from 184 to 186 of the main text.

FIG. R2. Comparison of the theoretical prediction, fitting and experimental data.

—Furthermore, how do you know you’re saturating the Cramer-Rao bound ? This metric seems, as a lower bound, is likely making your precision out to be better than it actually is, unless you can show that you’re indeed hitting that bound. How are you determining the data shown in Figure 1(f) ?

Response: We thank reviewer for this valuable question. The Cramer-Rao-lower-bound is the theoretically calculated best precision one can get using any possible estimators (i.e. localization algorithms in this case)[J. Opt. Soc. Am. A 33, B36 (2016)], it is widely used in the field of localization microscopy as a metric to compare different techniques [Phys. Rev. Lett., 124, 198104 (2020), Phys. Rev. Lett., 113, 133902 (2014)] although in practice it is often not saturated.

The localization precision in our experiment not only depends on the fundamental limits described by the CRLBs but also uncertainties from other aspects including ion movement, and vibration from the imaging system, camera readout noise, etc., therefore the CRLBs in our experiment is not saturated. Despite that, it can be approached by a time consuming maximum-likelihood estimation [Opt. Express 13, 10503 (2005), Opt. Express 26, 7965 (2018)] provided that the system is shot-noise limited and a complete library of calibrated PSFs is available.

The CRLB data in Fig.1f are theoretical calculations based on the Fisher information model described by Chao et al. [J. Opt. Soc. Am. A 33, B36-B57 (2016)], and the relevant parameters (e.g. NA, photon level, etc) are set to meet our experiment conditions. The purpose is to show that helical PSF carries more information in the axial direction compared to the conventional method of defocused Airy disk, thus, it's more suitable for 3D imaging of ions. We have modified the caption for Fig.1f and main text in line 93 to 96 to make this clearer.

–What do the lines in Fig. 2 (and Fig. 3) mean—are they fits ? How do they compare to what you expect analytically (i.e., how do you know that aberrations aren't causing you to misinterpret your data) ? What do the error bars mean ? The timestamps above Fig. 2(e) are sloppy and should be modified.

Response: Thanks for the questions. In Fig.2, the lines are indeed fittings. Due to the inaccessibility of vacuum chamber, our system has to be calibrated in experiment. Fittings are used to find out the relation between the driving voltage and the ion amplitude, and the relation between the PSF rotation angle and the axial displacement of the emitter. The calibration is done prior to the experiment and is then used for data processing. The process is system specific and the systematic aberrations can be accounted. We have added a comparison of the fitting to corresponding theoretical prediction in Fig.2e, where they show acceptable consistency.

In Fig.3, the solid lines are parameter fittings to the theoretical model in Eq.(3) and Eq.(4). This is regarded as an evidence that the 3D motion of the ion is following the modified forced vibration theory, confirming the experimental data for 3D motion is valid.

The error bar is the standard deviation of image fitting, the 3D coordinates of emitter are acquired by fitting the PSF image with a two-peak two-dimensional Gaussian model, followed by extracting the rotation angle. To make this clear, we added the notions for the error bar in the caption of Fig.2, where the description is “Error bars are standard deviations from image fitting...” (LOR14).

Thanks for the suggestion on timestamps, we have modified these timestamps, now they should fit the style of Fig.2.

–What is an “extra delay phase” ? Why is this important ?

Response: Thanks for the question. We feel sorry that we did not make it very clear in the main text. The extra

delay phase actually means the phase difference between the theoretical prediction and experimental measured curves in Fig.3d. This delayed phase results from the electric circuits of the system and can be determined from the fitting of experimental measured data. Therefore, we need to add an extra phase to our analytical model, which is used to fit our experimental data. To make the sentence clear, we rephrase the sentence in line 155 to be "..., which is finally employed in the trajectory analysis along z axis by introducing an extra delay phase (see supplement for details), ...".

—You mention being able to image an ion’s position in 3D in one snapshot, then you say that you can image phase transitions at a speed as fast as the camera’s frame rate. Don’t you need multiple images to determine phase transitions ? This is confusing as written. It also seems somewhat arbitrary with regards to what the authors call the zig and zag modes. Is this common parlance in the field ? I’m not sure here. If so, a citation to similar work that discusses this would be prudent.

Response: Thanks for these comments. (i) We now realized that this sentence can be misleading and have revised it to be more accurate in line 170 to line 172: “Since the ion coordinates can be deduced from a single snapshot at one moment, the proposed approach is capable of monitoring the 3D position dynamics during the phase transition at a speed as fast as the frame rate of the camera.” (ii) For the “zig-zag” mode, there is no clear difference between these two words, they are used to describe two conjugate configurations. Similarly to the words left and right, their definition is observer-dependent. “Zig-zag” modes are commonly used terms in the field of structural transition of trapped ions, to make this clear we have added the reference [Physics Letters A, 380, 2644 (2016)] in line 160 to help readers understand the meaning.

—The authors should comment on how tightly packed the imaged atoms/ions can be before the two helicoidal points of adjacent emitters overlap and can thus cause confusion.

Response: Thanks for the suggestions. The Coulomb repulsion in the ion trap results in an ion-ion spacing of over $5\ \mu\text{m}$ to $8\ \mu\text{m}$ in an ion chain. In our experiment with multiple ions, the two-lobe spacing of the helical PSF is around $5\ \mu\text{m}$ and we align the two-lobe direction perpendicular to the trap axis, so that the in-focus image of nearby ions do not overlap. Considering the rotation rate of the helical PSF is about 35° per $20\ \mu\text{m}$, the images of nearby ions would not overlap over the whole depth range of interest.

However, under extreme circumstances where more densely packed Coulomb crystals of complex 3D structures are present, the images could overlap and cause confusion. In such scenario, it has been shown in the field of super-resolution biomedical imaging that algorithm based on convolutional neural networks could be used to successfully decode information from the severe overlapping images, some typical demonstration can be found in reference [Nat. Methods 17, 734–740 (2020), Nat. Methods 18, 1082–1090 (2021)].

—I would like to see more details on how the phase mask is produced and implemented.

Response: Thank you for reminding us to add the details of the phase mask. The phase mask is important for readers who are not familiar with the imaging technology, now we have added the detailed process for generation of the phase mask both in main text in line 69 to 86 (LOR9, LOR10, LOR11) and in the supplement section IV (LOR22) and have commented on its possible implementation configurations (LOR24).

–Finally, while this paper presents some interesting work, but it is generally poorly written. This causes it to be confusing in parts. Extensive editing work is required before it will be acceptable for publication in a Nature journal, I believe.

Response: Thanks for valuable comment on the language problem. We have carefully revised this paper based on reviewers’ comments and used the English editing service to polish the language in this manuscript, we believe it has been improved now and is easier for readers to understand.

Reviewer #2 (Remarks to the Author):

The authors of the manuscript “Tracking the extensive three-dimensional motion of single atoms by engineered point spread function” devise and demonstrate a novel imaging method to determine the three-dimensional position of trapped ions (or other point-like scatters) under a microscope. Drawing on methods developed in other contexts, they implement a phase mask in the imaging path that transforms a Gaussian point spread function into a two-lobed structure, whose rotation encodes the axial position of the scatter. Using this technique, they demonstrate the three-dimensional tracking of a single ion undergoing periodic driven motion and the zig/zag orientation of a five-ion crystal.

The imaging method demonstrated here is not completely novel conceptually, but to my knowledge, this is the first implementation in the context of quantum sciences and imaging of trapped atoms. The applicability of the scheme in most typical ion trapping scenarios is not quite clear, in particular, because the photon number requirements are not discussed very clearly (see below).

Nevertheless, the use of the helical PSF in this context is novel and may lead to interesting applications in ion or neutral atom trapping. The manuscript is clear, cites prior work appropriately, and described the data in sufficient detail.

I recommend publication of the manuscript provided that the authors can address the questions below.

Response: Thanks for the appreciation and recommendation of this paper, the questions are addressed below, please check the detailed response.

There are two main questions that I feel are not discussed extensively enough in the main text.

1. Photon budget. As mentioned briefly in the methods, the resolution of any imaging method is ultimately limited by the number of detected photons. This fundamental limitation is not discussed at all in the main text and in my opinion, deserves a better exposition. In particular:

a. Looking at Figure 1F, the fundamental difference between naive Gaussian and the helical PSF methods is not very large, especially in the transverse directions. Does the helical PSF actually outperform traditional methods once the finite efficiencies of the phase mask are taken into account ?

Response: Thanks for valuable question. According to the Fisher information theory, the fundamental limitation of localization precision lies in the intensity field variation rate, i.e. $\frac{\partial I_{PSF}(z,h,v)}{\partial z}$, $\frac{\partial I_{PSF}(z,h,v)}{\partial h}$, $\frac{\partial I_{PSF}(z,h,v)}{\partial v}$, for estimation of emitter's 3D coordinates. On top of that, the localization precision scales with the total number of collected photons (Fig.R4), we have clarified this point in lines from 101 to 104, and lines from 209 to 211. Besides, a detailed comparison of various signal photon numbers have also been provided in the supplementary section V (LOR27).

We agree that the precision for naive Gaussian and the helical PSF methods are quite close in the transverse directions, as shown in Fig.1f. To make this point clearer, we have added the sentence “ ...In the lateral direction (h and z), double helical PSF does not show advantage over the defocus Airy disk.” in section V of the supplement.

However, in the direction of propagation, the localization performance is fundamentally limited by the way light field evolves, i.e. $\frac{\partial I_{PSF}}{\partial v}$. In the case of the naive Gaussian PSF, its variation rate is close to zero near the focal plane (Fig. R3). Little information of the ion's v position can be extracted from the Airy disk (between $-2\mu\text{m}$ and $2\mu\text{m}$ for Fig. R3). On the contrary, the double helical PSF yields constant rotation as light propagates, yielding a much better localization performance.

In this regard, the advantage still holds even though the helical PSF contains fewer photons. To make this point clearer, we have added the limitation discussion for the v direction in lines from 101 to 104, where the sentence is “The amount of Fisher information in the PSF image, regarding the ion's axial coordinate v , is positively correlated to its variation rate along this direction, i.e. $\frac{\partial I_{PSF}}{\partial v}$ [45]. The Airy disk yields no intensity variations along the optical axis direction near the focal plane, leading to its poor localization precision in v ”.

b. The data shown here is for very large photon numbers of 22k photons per atom. This number is quite atypical for “normal” ion trap experiments, where high-fidelity imaging can be performed with significantly fewer photons. Does the helical PSF advantage translate to lower photon number of several hundred in more typical experiments ?

Response: Thanks for the comment. 22k is quite a large number for state detection of trapped ions, but this photon number can be reached for precise determination of ion's position, since the photon rate can reach over 100k/s in our trapped ion experiment. To investigate the advantage of helical PSF, we have added the CRLB comparison between

FIG. R3. Experimental comparison of single-ion image stacks taken with conventional method of defocused Airy disk and double helical PSF near the focal plane. Images are taken under same imaging conditions with same photon counts.

FIG. R4. Comparison of helical PSF and conventional PSF using the CRLB metric with 500, 1500 and 2000 signal photons respectively.

conventional Gaussian PSF and the helical PSF for various signal photon levels in the section V of the supplement. As can be observed in Fig.R4, fewer signal photons results in reduced localization precision in both methods, however, the advantage of the helical PSF near the focal plane still holds. The amount of information possessed by the light field of the emitter’s depth is positively related to the variation rate of the PSF along the propagation direction. The conventional PSF (naive Gaussian) exhibits zero variation rate near the focal plane. We have added discussions in the supplementary section V to compare the performance of the two methods at various photon levels and the result is shown in Fig.S8. (LOR27)

2. Design of the helical PSF. It would be nice if the reader could build some intuition for the design of the helical point spread function and follow the considerations that went into the choice of this particular phase mask.

Response: Thanks for the suggestion. To give a clear choice for readers who may use this method, we have added the details of phase mask design in the supplementary section IV and provided detailed description on the design parameters for different purposes, we also listed some references for readers to choose other types of engineered PSFs and phase masks for their own research purpose, the sentence can found as “ In addition, other types of engineered PSFs may be employed in 3D ion imaging potentially. For example, the Airy-beam-based PSFs [Nat. Photon. 8, 302 (2014), Phys. Rev. Lett. 124, 198104 (2020)] offer the largest depth range with a compromised precision and

the Tetrapod PSF is demonstrated in extremely high emitter density [Nat. Methods, 17, 734–740 (2020)]. Several review articles can be referred to when choosing between different phase masks and corresponding engineered PSFs [Biophysical Reviews 12, 1303 (2020), APL Photonics 4, 060901 (2019), Reports on Progress in Physics 78, 124601 (2015)]. We look forward to the application of more PSF engineering techniques to be exploited in atomic-scale detect”. We believe readers now can follow the considerations and design their phase mask they need.

a. What is this phase mask, e.g. which Zernike polynomials contribute to the image shown in Figure 2c ?

Response: Thanks for the valuable comment. Following the reviewer’s suggestion, we have decomposed the helical phase mask to the first 55 Zernike polynomials (i.e. the first 9 orders) and the results are illustrated in Fig.R5. One can observe that many terms contribute to the helical phase mask including the vertical astigmatism, horizontal coma, etc. It can be expected that Zernikes of very high orders will also make a contribution, and the mask can not be represented by a limited number of Zernike polynomials. The results can be interpreted that the phase singularities and 2π phase jumps in the helical phase mask (which are essential for conveying orbital angular momentum to the light field [Opt. Express, 20(24), 26681 (2012)]) can not be well described by the Zernike polynomials, because no Zernike polynomial term shows such properties.

FIG. R5. Decomposing the helical phase mask to the first 9 orders of Zernike polynomials.

b. How do we understand the difference in transverse performance between the Gaussian and helical PSF displayed in Figure 1F ? I assume the comparison is carried out at a fixed number of detected photons, but please clarify.

Response: Thanks for the comment. The difference in transverse performance between the two PSFs can be under-

stood in two aspects. (i) Near the focal plane, the conventional Gaussian PSF yields the most compact and brightest intensity profile and the largest variation in presence of lateral shift ($\frac{\partial I_{PSF}}{\partial z}$ or $\frac{\partial I_{PSF}}{\partial h}$), therefore it gives the best performance in the lateral direction. (ii) On the other hand, the helical PSF exhibits a larger depth range compared to the conventional Gaussian PSF, thus, its CRLB in the lateral direction degrades slower than the rapid expanding Airy disk as defocus increases. The theoretical comparison in Fig.1f is implemented under the assumption of fixed detected photons. To make this point clear, we have added this simulation condition the caption of Fig.1f and the sentence is “ Theoretical comparison between the localization precision of the helical PSF (red curves) and defocused Airy disk (green curves) methods estimated using the Cramer-Rao bound metric under same imaging conditions (0.4 NA, 22k signal photons per ion and 10 background photons per pixel)”.

One could determine the position of an atom simply from the center of mass of all detected photons. In this case, I would imagine that at fixed photon number and resolution, a Gaussian PSF is the best choice because of its minimal spatial size.

Response: Thanks for the valuable comment. We totally agree with reviewer’s opinion that the Gaussian is the best choice for the lateral position measurement. However, it is not a good choice when we have to consider the motion along the optical axis. As indicated in Fig.R3, we cannot determine the displacement along the optical axis with very high precision due to the fact that the point size is not sensitive to the displacement v . Please note that the images in Fig.R3 is taken with same photon counts. To make a clear description of the lateral precision of these two PSFs, we make a comparison for various experimental conditions (Fig.S8) and added the sentence “ ...In the lateral direction (h and z), double helical PSF does not show advantage over the defocused Airy disk.” in section V of the supplement.

If, on the other hand, the spatial position is determined from performing a more elaborate fit on the PSF, having sharper features (or more lobes) in the PSF might be preferable. Would having more lobes in the PSF improve the transverse resolution even further ?

Response: Thanks for the comment on other PSFs. According to the theory of Fisher information, having sharper features in the image (i.e. a larger $\frac{\partial I_{PSF}}{\partial h}$, or $\frac{\partial I_{PSF}}{\partial z}$) indeed gives more information in the transverse direction. Therefore, if PSFs have more lobes we can possibly have better measurement precision in the transverse direction (h or z). However, the fluorescence of a trapped ions is almost a constant in the experiment, more lobes indicates that the photons in each lobe will be less, and the measurement precision for each lobe will be worse, we cannot predict the final transverse resolution will be better in this case.

Please comment on the strategy of choosing a particular phase mask.

Response: Thanks for the comment. The choice of phase mask should be made according to the purpose of each experiment. We have added the following text in supplementary section V:“ In addition, other types of engineered PSFs may be employed in 3D ion imaging potentially. For example, the Airy-beam-based PSFs [7, 8] offer the largest

depth range with a compromised precision and the Tetrapod PSF is demonstrated in extremely high emitter density [9, 10]. Several review articles can be referred to when choosing between different phase masks and corresponding engineered PSFs [11–13]. We look forward to the application of more PSF engineering techniques to be exploited in atomic-scale detect”

c. Is there experimental evidence that the theoretical improvement in resolution reported in Figure 1F is actually observed in the experiment ?

FIG. R6. Experimental comparison of the axial sensitivity by calculating the helical rotation and the Airy-disk full width at half maximum (FWHM).

Response: Thanks for the comment. The main advantage of double helical PSF over the conventional method of defocused Airy disk is in the optical axis direction v . To demonstrate this advantage, we took another set of experimental data, the results are shown in Fig.R6. It can be seen that the helical PSF exhibits a relatively consistent rotation while the Airy disk yields a inconsistent diameter change. In particular, the Airy disk diameter changes slowly near the focus, indicating a poor localization precision. Please note that the data processing is done by following methods: for the helical PSF the two lobes are fitted to a two-peak 2D Gaussian function and calculate their mutual rotations while the Airy disk PSF is fitted using a single-peak 2D Gaussian function to calculate its full width at half maximum (FWHM). Above experimental results have been added to Fig.S7 and the following text “It can be seen that the helical PSF keeps a consistent rotation rate, leading to a consistent depth sensitivity over the range of $-25 \mu\text{m}$ to $25 \mu\text{m}$; while the Airy disk diameter expands non-linearly and exhibits near-zero change rate near the focal plane” has been added to the section V of supplement.

Minor points

A. How is the phase mask implemented physically ? What efficiency is observed for the physical implementation ? How does that feed into the question of the photon budget mentioned above ?

Response: Thanks for the questions. Typically, the phase mask can be implemented using a spatial light modulator or a laser-written refractive glass substrate. The SLM modulates only one polarization of the light (50%) and exhibits a limited fill factor (typically around 95%), while the laser-written refractive glass substrate can provide near-unity photon efficiency. In our experiment, we measured our refractive mask to have an optical throughput of 92%; the overall efficiency of a LC-SLM (Santec SLM-200) to be around 42%, as can be seen from Fig.R7. Considering all the optical elements in the imaging system except the phase mask, we have photon budget of about 114k/s for a single ion. Even with SLM phase mask included, we can still have a photon count rate of around 47.9k/s; therefore the photon budget for ion imaging in previous discussions can be reached.

To make this clearer, we have added description on the phase mask implementation and corresponding efficiency in supplement section IV: “...The phase mask maybe implemented in several ways, such as diffractive liquid-crystal SLM in a folded $4f$ configuration or refractive laser-written glass substrate in an inline configuration, SLMs modulate one direction of polarization (50%) and yields a limited fill factor (around 90%), yielding an overall efficiency of around 45%. On the contrary, a glass phase mask can reach an efficiency of over 90% but at the expense of flexibility...”

FIG. R7. Measurement of the SLM efficiency, modulated light is separated with the unmodulated light spatially by adding a blazed grating to the helical phase mask. The overall photon efficiency of the SLM was measured to be around 42%.

B. Line 60. Please specify how short the lifetime is and at what rate photons can be collected.

Response: We Thank reviewer for these suggestions. The lifetime of transition $S_{1/2} \leftrightarrow P_{1/2}$ is 7.1 ns and the detected photon count rate is about 114 k/s by PMT in the experiment, we have added this revision to line 61 to 63 in the main text and a reference [Phys. Rev. Lett. 70, 3213 (1993)] about the lifetime of state $P_{1/2}$ has also been added (LOR7).

C. Line 78. How do the authors arrive at the “order of magnitude better range-to-precision ratio” ? The doubling of the range at roughly similar precision (except near the focus) would suggest this improvement is about a factor of two. If the focus is included, where the imprecision of the Gaussian method diverges, the factor would be infinity. Why one order of magnitude ?

Response: We are thankful that the reviewer pointed this issue out, it was indeed lack of clarity. By ‘range-to-precision’ we meant the range divided by the average precision over the range, with the focal area included. By this definition, the helical PSF shows more than ‘one-order-of-magnitude’ advantage on the range of interest. To make this statement clearer, we have added definition for ‘range-to-precision’ in the main text in line 100 , now the description is “...an-order-of-magnitude better range-to-precision ratio, which is defined as the detected range divided by the average precision over the range in the vertical direction” (LOR12).

D. Line 113. The coefficients beta and c are not defined anywhere in the main text.

Response: Thanks for the comment. To make the coefficients clear in the main text, we have added the linear fitting function ($v(\theta) = \beta\theta + c$) in line 139, the definition of β and c is described as “ Then the fitting constants β, c that reveals the relation between the rotation angle θ and vertical displacement v can be found in the linear model $v(\theta) = \beta\theta + c$, the parameters are calibrated as $\beta = \frac{180}{\pi} \times (0.57 \pm 0.02) \mu\text{m}/\text{rad}$ and $c = -0.22 \pm 0.32 \mu\text{m}$ in Fig.2f.” (LOR15)

Reviewer #3 (Remarks to the Author):

This paper demonstrates a new approach to performing 3D imaging of single ions using a phase-mask to transform the point spread function from an Airy function to a helical beam profile that rotates proportional to the change in depth of focus, overcoming a major limitation of previous approaches to 3D imaging using either adjustable focal length lenses or moving optics, or by inferring from blurring and defocus of the Airy spots. To demonstrate the effect the authors show real-time imaging of a 5 ion crystal transforming between two different zig-zag configurations, and demonstrate a theoretical resolution of 20nm in a 20 μm depth of focus. These results provide an interesting approach to performing 3D imaging which is relevant to work on both trapped ion systems and emerging neutral atom tweezer arrays, making this an important and impactful paper.

Response: Thanks very much for the positive comment on our work.

However, in current form, there is not sufficient detail to allow another group to easily adopt and replicate this data which makes it hard to recommend for publication in Nature Comms in the current form despite the novelty of the results. For example, there are two pages of supplementary material explaining the ion trap setup and almost no detail at all about the design and optimization or fabrication of the phase-mask that underpins the results which

for most readers is the crucial part of the paper. Additionally, there is a lack of discussion around how this readout scheme can be used in the situation of high-fidelity readout and state detection which for many applications of 3D imaging is required.

Response: Thanks for the valuable comment on the fabrication details of phase mask. We are very sorry that the details for design the phase mask is absent in the current form. To make the design and optimization of helical phase mask clear to readers, we have added a new section “ Generation and implementation of the helical phase mask ” in supplement (section IV), where the design flowchart, optimization function, implement material are included. Now we believe readers can design and optimize phase mask for their own purpose. For the readout scheme, we are sorry that we did not tell readers how to use it. Actually this read out is compatible with the state-of-the-art imaging systems of cold atoms and ions. When the 3D information of a system is wanted, one only needs to modify the imaging system to a $4f$ relay system and place the phase mask to the re-imaged pupil plane of the system as shown in Fig. 1a of the main text. To make this point clear, we have also added the sentence how to use this detection method in lines from 177 to 178, now the description is “...both trapped ions and neutral atoms, the implementation of the detection scheme only requires a $4f$ relay system and an appropriate phase mask.”

I propose the authors resubmit a revised manuscript addressing the main criticisms below to allow consideration of whether this is sufficiently detailed for publication in Nature Comms.

Response: Thanks for the suggestion. We also realized that the current manuscript did not provide enough details for readers who are not very familiar to the engineered point-spread function. To make the paper easy to be understood by more readers, we have added details of the design and optimization of the phase mask, discussion of the photon budget, precision variations as a result of different photons and NAs to the supplement as two sections. Also we make a major revision to the main text, all modifications are marked in red, now we believe the manuscript has enough details for Nature Comms.

(1) As mentioned above the key focus of this paper is on use of the phase-mask to transform the PSF, but there just is insufficient detail supplied. On page 2 it simply states the phase mask was “optimized using the iterative Fourier transform algorithm to reduce the number of diffraction edges.” but this requires more detail so a reader could design a suitable phase mask from scratch e.g. what is the cost function used in the minimisation process? How was the initial phase-mask chosen to choose the mapping from focus to angle? There is also no detail about mask construction - how is the experimental phase mask fabricated, from what materials and with what process? What is the tolerance on the mask compared to the theoretical design?

Response: Thank you for the comment on the phase mask. We are sorry that we did not provide enough information about the design and optimization of helical phase mask. To help the readers to design their own phase mask, we have provided a general PSF design flow chart in the section IV of the supplement with a detailed description. The initial phase mask was chosen depending on the specific working conditions: in low-photon budget experiment here, we chose the parameters “ $L = 4$ and $\eta = 0.9$ for a high peak confinement and fast rotation rate.” The rotation rate

can be calculated by numerical calculations using the imaging model added in line 68 to line 85, so as to check on the mapping from focus to angle.

We implemented the mask in two ways in the experiment: (i) via laser written on UV fused silica, tolerance is about $\lambda/4$. (ii) via liquid-crystal spatial light modulator, tolerance can be calibrated below $\lambda/20$ [Opt. Lett. 48, 5-8 (2023)]. Above details about phase mask has been added to the section IV of supplement (LOR22 and LOR24).

(2) For performing high fidelity imaging, the aim is to obtain high collection efficiency onto a small group of pixels. For the helical PSF, how efficient is the transformation, and how would the fidelity of detection evolve comparing the Airy and helical PSF accounting for the change in number of pixels and larger contribution to background counts of spreading the signal over a larger number of pixels.

Response: Thanks for the comment. To figure out the evolution of detection fidelity, we have investigated numerically the evolution of the two types of light field along the optical axis. As shown in Fig. R8(a-b), the conventional PSF focuses tightly near the focus but expands dramatically out of the Rayleigh range, while the helical PSF yields relatively larger footprint but keeps compact over a larger depth range. This is also investigated quantitatively in two aspects: Fig.R8(c) shows the FWHM of the main lobe of the both PSFs (simulated with same photon count) and how it changes as light field evolves. Fig.R8(d) shows a comparison of the number of photons in the main lobes between the conventional PSF and the helical PSF (two-lobe summation). From the simulations, it can be observed that the helical PSF yields a transformation efficiency of around 50% near the focal plane which slowly degrades as the defocus increases, while the photons of conventional PSF spread much faster.

FIG. R8. Illustration of 3D light field for (a) conventional PSF and (b) helical PSF. (c) Main lobe FWHM evolution along the optical axis. (d) Photon counts in the main lobes evolving along the optical axis, counting only photons within a 10-pixel-diameter circle indicated in the above.

(3) The authors quote a resolution of 20 nm at 20 μm depth of focus. How do these numbers scale with e.g. NA - if you increase the NA from 0.4 to 0.5 how does the depth of focus and resolution change- is it a simple relationship?

Response: Thanks for the question. We have added simulations on the CRLB comparison for different NAs, which provides an insight into how the performance at all directions scales with NA, and the simulation results are shown in Fig.R9. It can be observed that a higher NA leads to a shallower depth range and better precision in all directions for both techniques. We have included these results together with a brief discussion in the supplementary section V: “...Fig.S8(a-c) compare the theoretical precision for different NAs at 0.3, 0.4 and 0.5 respectively. As one can observe, localization performance for both methods in z, h, v improves as NA increases on a shallower depth range. Of particular interest, the helical PSF shows a more consistent CRLB over the entire depth which compares favorably to the method of defocus Airy disk regardless of NA...”.

FIG. R9. CRLB comparison between the helical PSF and the defocused Airy disk for numerical aperture of 0.3, 0.4 and 0.5. Plots from left to right show theoretical precision for transverse coordinates z , h and vertical coordinate v .

(4) For many of the example applications of imaging single atoms and ions another important factor is cross talk. How close can I resolve two objects at different distances? Is there a trade-off between the design of the helical and the resulting resolution that can be obtained? Ideally the figures should be clearly labelled with distance in μm to aid in resolving the PSF size scale relative to atom spacing.

Response: Thanks for the comment. The two-lobe spacing of the helical PSF in the ion chain is around $5\ \mu\text{m}$, so ions with spacing larger than two-lobe spacing can be resolved considering an ion-chain configuration. The Coulomb repulsion in our ion trap results in an ion-ion spacing of $5\ \mu\text{m}$ to $8\ \mu\text{m}$. Considering that we align the two-lobe direction perpendicular to the trap axis and the rotation rate is only 35° per $20\ \mu\text{m}$, cross talk is absent in our experiment. Note that one can alter the two-lobe spacing by changing the phase mask parameter L , a smaller L leads to a smaller two-lobe spacing and thus better resolution in a given system, therefore one needs to balance the resolution and design parameters based on experimental conditions (the discussion can found in the section IV of the supplement).

To give a clear scale for the PSF size and trapped ions spacing, we added scale bars for all figures in the main text.

(5) In the abstract and manuscript the expression “a thin detection range” is used and this is not sufficiently clear - I believe what is meant is a shallow depth of focus but clarity is required in this phrasing.

Response: Thanks for the suggestion. To make this clear, we have changed the word "thin" to "shallow", now the sentence is "the current single-atom-resolved 3D imaging methods are limited to static circumstances or a shallow detection range." (LOR3).

(6) In the introduction where "3D methods for imaging" are discussed it would be helpful for the general reader to appropriately cite prior work performing 3D readout using movable lenses (e.g. <https://doi.org/10.1038/nphys645>), focus adjustable lenses (ref 22) and using an SLM to re-map the 3D space onto a 2D plane (<https://doi.org/10.1364/OE.415805>)

Response: We thank reviewer for these valuable comments on the references. We have added the citation of [Nature Physics 3, 556 (2007) and Opt. Express, 3, 4082 (2021)] in the sentence "There also exist a few 3D imaging techniques that can determine the 3D positions of single atoms [9–15],..." (LOR4).

(7) Whilst the paper is generally well written, there remain many instances where articles are incorrectly used or the meaning is unclear e.g. in the abstract "acquisition of ion's 3D position" should be "the ion's position", and on line 33 "In direction, the position..." - what direction is being referred to? Please carefully review the grammar.

Response: We thank reviewer for the suggestion on the writing. We have carefully checked the whole manuscript and made revision for the sentences and phrases, all the modifications are marked in red, now the whole manuscript should be much better. Hopefully the manuscript now meets the criteria of the Nature Communications.

(8) An equation is given to describe motion (eq 2) with no citation - how is this derived?

Response: Thanks for the question. The Eq. (2) is derived by modifying the current forced vibration equation according to the trapped ion potential, and we have added 2 citations [Journal of Sound and Vibration 330, 4313 (2011), Scientific report, 13, 6507 (2011)] we referenced in line 108, now the sentence is "To track the large motion range of a single ion, we drive the ion with extra RF electrical signals to excite forced vibrations [47, 48]".

II. LIST OF REVISIONS (LOR)

LOR1: the word "atoms" is changed to "ions" in the title.

LOR2: In the abstract, the English wording and sentences have been polished and the word "atom" is revised to be "ion" where an ion is referred.

LOR3: the word "thin" has been changed to "shallow" for better expression.

LOR4: More references on the static 3D imaging methods have been added to line 24, and references on stacking techniques have been added to line 39.

LOR5: A PhD thesis has been added in line 55 to give a reference for the readers including the information on the trapping theory, laser systems, imaging system, quantum state operations, etc.

LOR6: Fig.1f has been modified to be clearer and more relevant calculations on the CRLBs have been added to supplementary material.

LOR7: A more detailed description on the lifetime and photon count rate have been added to lines from 61 to 63.

LOR8: More details of engineered PSFs has been provided with several references added from line 65 to 68.

LOR9: We added more description on the imaging model to lines from 68 to 74.

LOR10: A discussion about the possible implementation has been added to lines from 75 to 76.

LOR11: More details regarding the generation of the specific helical phase mask and the iterative optimization algorithm have been added to lines from 77 to 86.

LOR12: The definition of “range-to-precision ratio” has been added to line 100.

LOR13: The fundamental limitation on estimating the ion’s coordinates has been discussed in lines from 101 to 104.

LOR14: We added a sub figure Fig.2f as a comparison between the theoretical imaging model and experimental data regarding the rotation angle of the PSF as a function of the vertical displacement. Also the description of error bars and scale bar in Fig.2e have been added.

LOR15: The linear model $v(\theta) = \beta\theta + c$ has been added to line 139 to give a clear definition of constant parameters β and c .

LOR16: A discussion on the possible aberrations in the experiments and importance of calibration has been added in lines from line 141 to 145.

LOR17: We have rephrased the inaccurate sentence on the imaging speed of our method in lines from 170 to 172.

LOR18: A sentence about how to implement the imaging method in current system is added in line 177 and 178.

LOR19: Future improvement has been discussed in lines from 184 to 186 to give a guide for better calibration of imaging systems.

LOR20: A discussion on the relationship between the localization precision and the number of detected photons has been added from line 209 to 211.

LOR21: We have revised the symbol ϕ_1 and ϕ_2 to be γ_1 and γ_2 to avoid conflicts in the usage of symbols.

LOR22: We have provided detailed description on the generation and implementation of the helical phase mask. And a detailed flowchart illustrating the adapted Gerchberg-Saxton algorithm has been added as a new section IV of the supplement.

LOR:23 A numerical comparison of the vertical localization precision using various phase mask parameters has also been discussed in the section IV of the supplement.

LOR:24 A detailed discussion on the implementation of the phase mask using liquid crystal SLM or the laser-written glass substrate has been added in the section IV of the supplement, where the optical efficiency of both methods has also been discussed.

LOR:25 The experimental demonstration showing the advantage of the helical PSF over the conventional defocused Airy disk method has been added to the section V of the supplement.

LOR26: Theoretical localization precision comparison between the helical and defocused Airy disk under the condition of different numerical apertures has been added in the section V of the supplement.

LOR27: Theoretical localization precision comparison between the helical and defocused Airy disk under the condition of different photons has been added in the section V of the supplement.

LOR28: Discussion on the possible usage of other types of engineered PSF in the field of single-atom-resolved imaging has also been provided in the section V of the supplement.

REVIEWER COMMENTS

Reviewer #1 (Remarks to the Author):

I would like to thank the authors for returning the manuscript; in its updated form it is much more informative.

There are a few points that still need to be made before I support publication, although they are minor:

First, the authors should cite this extremely relevant paper that was recently published in PRA: <https://journals.aps.org/pr/abstract/10.1103/PhysRevA.109.033304>, which, to the best of my knowledge, is the first demonstration of helicoidal PSFs for atom imaging, with an extensive analysis of how aberrations affect the data. The authors' paper is still novel, however, in that it tracks dynamics (and their data is much cleaner).

One point to be made is that Figs. 1b and 1c in the above paper seem to conflict with Fig. 1f in the authors' paper in that the Fisher information of the double-helix PSF was larger than the standard PSF. This may be due to the differences in the chosen PSF engineered and differences in how the precision was retrieved from the Fisher information (and Cramer-Rao bound), but should be noted and could potentially be discussed by the authors. I note also that it seems that the authors did include shot noise in their model, while the other paper does not, so this could explain things to some degree (although perhaps not all of the factor-of-two difference seen in the PRA paper).

On a related note, it's also worth describing how the authors got to the statement that "[their] PSF design can possibly yield a 3D localization precision better than 20 nm over a depth range of 20 μm ". Basically, how did you back out a precision from the CRLB?

Atoms and ions are now differentiated in this work, but "atomic" is still used where "ionic" should be, in places.

The language still suffers throughout the manuscript. In particular, the methods section (under "Fluorescence collection for a single ion") should be revised for clarity.

In the supplement, "field" is misspelled in "sampled light filed at K planes".

Reviewer #2 (Remarks to the Author):

The authors have substantially revised the manuscript and added new experimental data showing the performance of their imaging scheme. Most of my criticism has been addressed and I see the manuscript as a solid description of a novel imaging method that will trigger substantial interest in the field.

What I still find lacking is a description of the physics of the helical phase mask. My request to describe the phase profile of the mask was not aimed at a list of numerical Zernike coefficients, but at an intuitive description of how the method works. The authors now give the mathematical form of their initial phase mask and provide a flow chart for their numerical optimization scheme thereof, but very little is said about the physics of their engineered PSF.

The main angle of the paper is to bring a method from biological microscopy to the world of quantum sciences and to educate that community about the benefits and drawbacks of engineered point spread functions. Given this almost educational nature of the paper, I find it very surprising that the authors

do not make any effort to give the reader an intuitive understanding of the phase mask. Its design principles can be conveyed very clearly in a few lines, as demonstrated for example Opt. Express 22, 4029-4037 (2014). Why not describe the physics of the mask and educate the reader on its principles? A short discussion here could, in my view, significantly increase the impact of the paper.

The issue above should not preclude the paper from acceptance and I don't need to see it again before publication. However, I would strongly encourage the authors to help their readers by describing some intuition for helical PSFs.

Reviewer #3 (Remarks to the Author):

Following careful reading of the manuscript and the detailed response to the referee comments I am satisfied the authors have addressed all major concerns and the revised manuscript is now suitable for acceptance in Nature Comms. This is very nice demonstration of the ability to use a modified PSF for depth imaging of trapped ions, and will be relevant to a range of different systems using trapped ions or neutral atoms across a range of different quantum technologies.

I. RESPONSE LETTER TO REVIEWERS (NCOMMS-23-50566-A)

Reviewer #1 (Remarks to the Author):

I would like to thank the authors for returning the manuscript; in its updated form it is much more informative. There are a few points that still need to be made before I support publication, although they are minor:

Response: We thank the reviewer for thorough review, we have now addressed all the raised points, please find the details below.

First, the authors should cite this extremely relevant paper that was recently published in PRA: <https://journals.aps.org/pr/abstract/10.1103/PhysRevA.109.033304>, which, to the best of my knowledge, is the first demonstration of helicoidal PSFs for atom imaging, with an extensive analysis of how aberrations affect the data. The authors' paper is still novel, however, in that it tracks dynamics (and their data is much cleaner).

Response: Thank you for pointing out this work in neutral atom systems. We agree that this paper is of significant relevance to our paper and it is also a new 3D imaging method for the static neutral atoms. As the reviewer pointed out, this work is for static imaging of neutral atoms, while our work is for imaging dynamics. Therefore, we cited this work in the introduction part (line 24, reference number 16) where we discuss the existing 3D imaging methods for static atom systems. Additionally, we noticed that our first submission date (20th October 2023) to Nature Communications was actually BEFORE their first submission date (8th December 2023) of their work to arXiv and PRA.

One point to be made is that Figs. 1b and 1c in the above paper seem to conflict with Fig. 1f in the authors' paper in that the Fisher information of the double-helix PSF was larger than the standard PSF. This may be due to the differences in the chosen PSF engineered and differences in how the precision was retrieved from the Fisher information (and Cramer-Rao bound), but should be noted and could potentially be discussed by the authors. I note also that it seems that the authors did include shot noise in their model, while the other paper does not, so this could explain things to some degree (although perhaps not all of the factor-of-two difference seen in the PRA paper).

Response: We are sorry that the last submission was still confusing. In Fig. 1f of our manuscript we plot the theoretical localization precision (i.e. $\sigma_h = \sqrt{CRLB_h}$, $\sigma_z = \sqrt{CRLB_z}$, $\sigma_v = \sqrt{CRLB_v}$) predicted by the Cramer-Rao lower bound (i.e. the inverse of the Fisher information $CRLB_h = 1/I_{hh}^{-1}$, $CRLB_z = 1/I_{zz}^{-1}$, $CRLB_v = 1/I_{vv}^{-1}$). On the contrary, in the PRA paper, the Fisher information itself ($I_{ii}, i = h, v, z$) is plotted; therefore, the double helical PSF shows a greater magnitude near the focal plane in the PRA paper while it shows a smaller magnitude in our manuscript. Besides, as the reviewer has pointed out, this simulation varies with the chosen PSF type, relevant design parameters and also the noise model. To make the definition of CRLB clearer, we have modified in line 94 to 97: "The CRLB, being the inverse of the Fisher information, represents the best/ smallest possible variance of the parameters that can be reached when estimating the position of an ion, regardless of the estimator. Thus, the square root of the CRLB indicates the theoretically best localization precision."

On a related note, it's also worth describing how the authors got to the statement that "[their] PSF design can possibly yield a 3D localization precision better than 20 nm over a depth range of 20 μm ". Basically, how did you back out a precision from the CRLB?

Response: Thanks for the question, we are sorry that this conclusion regarding the theoretical localization precision in the previous version was unclear. The localization of point emitters can be treated as a parameter estimation problem for the spatial coordinates of h , z , v . According to the Cramer-Rao inequality, "...The CRLB, being the inverse of the Fisher information, represents the best/ smallest possible variance of the parameters that can be reached when estimating the position of an ion, regardless of the estimator[Nano Lett. 10(1): 211–218 (2010)]. Thus, the square root of the CRLB indicates the theoretically best localization precision..." These detailed description has been added in lines 94-97 to explain how the CRLB can back out the theoretical localization precision.

In addition, it has been shown this theoretical precision can be approached via the commonly-used maximum-likelihood estimator [Opt. Express 17, 23352–23373 (2009), APL Photon. 4, 060901 (2019)]; We have performed the following simulation as a further demonstration: 100 noisy images are generated for each v position based on the imaging model described by Eq.(1) and Eq.(2) in the main text; model parameters include the h z v coordinates and the signal and background photon counts. Using the Maximum likelihood function as an optimization objective following [Phys. Rev. Lett. 133902 (2014)], we showed that the standard variation for the v coordinate does approach the theoretical precision predicted by CRLB (Fig.R1).

FIG. R1. Numerical demonstration of the Maximum-likelihood estimator under the imaging condition of 22k signal photons and 10 background photons per pixel. For each v position, 100 images are generated and with Poisson noise added.

Atoms and ions are now differentiated in this work, but "atomic" is still used where "ionic" should be, in places.

Response: Thanks for pointing this issue out. We have checked through the manuscript and considered the usage of the word "atomic" carefully, and we did find three more places that are still ambiguous. We have revised in line 27: "...some examples include: observing the spatial transportation and quantum tunneling of ion qubits between different ion chips..." and in line 180: "...which has never been observed previously in cold ion or atom systems..."

and in line 196: “...the detection of the quantum tunneling on ion chips and in 3D optical lattices...”

The language still suffers throughout the manuscript. In particular, the methods section (under “Fluorescence collection for a single ion”) should be revised for clarity.

Response: We are sorry that there are still language issues in the manuscript. We have used the Springer Nature Language Editing service to help polish the English in the manuscript and made a substantial revision to the method section “Fluorescence collection for a single ion”. We hope that it now could meet the publication standard for Nature Communications. All changes are marked in red in the main text.

In the supplement, “field” is misspelled in “sampled light filed at K planes”.

Response: Thank you for pointing this out, we have now corrected this typo in the Fig. S5, and have checked all the figures and the text carefully for any existing typos.

Reviewer #2 (Remarks to the Author):

The authors have substantially revised the manuscript and added new experimental data showing the performance of their imaging scheme. Most of my criticism has been addressed and I see the manuscript as a solid description of a novel imaging method that will trigger substantial interest in the field.

What I still find lacking is a description of the physics of the helical phase mask. My request to describe the phase profile of the mask was not aimed at a list of numerical Zernike coefficients, but at an intuitive description of how the methods works. The authors now give the mathematical form of their initial phase mask and provide a flow chart for their numerical optimization scheme thereof, but very little is said about the physics of their engineered PSF.

The main angle of the paper is to bring a method from biological microscopy to the world of quantum sciences and to educate that community about the benefits and drawbacks of engineered point spread functions. Given this almost educational nature of the paper, I find it very surprising that the authors do not make any effort to give the reader an intuitive understanding of the phase mask. Its design principles can be conveyed very clearly in a few lines, as demonstrated for example Opt. Express 22, 4029-4037 (2014). Why not describe the physics of the mask and educate the reader on its principles? A short discussion here could, in my view, significantly increase the impact of the paper.

Response: Thanks for the appreciation and valuable comment, we have now realized your intention and have added more intuitive description on the physics of both the double helical phase mask and the light field evolution, which would be useful for broader readers to establish a picture of how this engineered PSF works. We have added in lines from 46 to 47: “...the ion’s position along the optical axis of the objective is encoded in the orbital-momentum-induced rotation of the two closely spaced PSF lobes...” and in lines from 86 to 90: “...When such a phase mask is implemented,

a typical Airy disk image (Fig. 1d) of the single ion is decomposed into two separated lobes, and each corresponds to half of the phase modulation pattern in Fig. 1c. This phase modulation introduces orbital angular momentum to the light field, leading to light field rotation with propagation: When the emitter is displaced in the v direction from the focal plane, the additional phase term (i.e., $D(\rho, v)$ in Eq. (1)) results in a continuous rotation of the two bright lobes in the PSF around their mutual center (Fig. 1e)...”

The issue above should not preclude the paper from acceptance and I don't need to see it again before publication. However, I would strongly encourage the authors to help their readers by describing some intuition for helical PSFs.

Response: Thanks again for the appreciation and recommendation of this paper, we believe the manuscript will benefit from this revision for capturing the broader interests of non-expert readers.

Reviewer #3 (Remarks to the Author):

Following careful reading of the manuscript and the detailed response to the referee comments I am satisfied the authors have addressed all major concerns and the revised manuscript is now suitable for acceptance in Nature Comms. This is very nice demonstration of the ability to use a modified PSF for depth imaging of trapped ions, and will be relevant to a range of different systems using trapped ions or neutral atoms across a range of different quantum technologies.

Response: Thank you very much for the recommendation of our work, we appreciate the thorough review of the previous round which has helped improve the manuscript significantly.

II. LIST OF REVISIONS (LOR)

LOR1: Explanation for the relationship of CRLB, Fisher information and localization precision have been added to lines from 94 to 97.

LOR2: The citation for PRA paper has been added to the line 24.

LOR3: Physical picture of the helical PSF and phase mask has been added to line 47 and lines from 86 to 90.

LOR4: English polishing for the whole manuscript by using Spring Nature editing service.

LOR5: The word “atomic” is modified to “ion” in line 27, 180 and 196.

REVIEWERS' COMMENTS

Reviewer #1 (Remarks to the Author):

Thanks to the authors for their response. I am happy to see the paper accepted.